# Shoreline Response to a Sandy Nourishment in a Wave-Dominated Coast Using Video Monitoring

**Catarina Jóia Santos [1],\*, Umberto Andriolo [2]** and **José C. Ferreira [1]**

[1]  Department of Environmental Sciences and Engineering, NOVA School of Science and Technology, NOVA University Lisbon/MARE-Marine and Environmental Sciences, Campus da Caparica, 2829-516 Caparica, Portugal; jcrf@fct.unl.pt
[2]  INESC Coimbra, Department of Electrical and Computer Engineering, Rua Sílvio Lima, Polo II, 3030-290 Coimbra, Portugal; uandriolo@mat.uc.pt
\*  Correspondence: csj.santos@campus.fct.unl.pt

**Abstract:** Beach nourishment is a soft engineering intervention that supplies sand to the shore, to increase the beach recreational area and to decrease coastal vulnerability to erosion. This study presents the preliminary evaluation of nourishment works performed at the high-energy wave-dominated Portuguese coast. The shoreline was adopted as a proxy to study beach evolution in response to nourishment and to wave forcing. To achieve this aim, images collected by a video monitoring system were used. A nourishment calendar was drawn up based on video screening, highlighting the different zones and phases where the works took place. Over the six-month monitoring period, a total amount of 25 video-derived shorelines were detected by both manual and automated procedures on video imagery. Nourishment works, realized in summer, enlarged the emerged beach extension by about 90 m on average. During winter, the shoreline retreated about 50 m due to wave forcing. Spatial analysis showed that the northern beach sector was more vulnerable and subject to erosion, as it is the downdrift side of the groin.

**Keywords:** beach; nearshore; remote sensing; erosion; sand

## 1. Introduction

Beach nourishment is a soft engineering intervention that consists of the injection and placement of sediments into the beach system, to extend, widen and elevate the subaerial beach [1–3]. The sediments added to the beach reduce the vulnerability to storms and enhance the wave energy dissipation [4]. Additionally, beach nourishment increases the recreational beach area, leading to positive consequences in tourism [5,6]. In order to evaluate the effectiveness of nourishment works, it is essential to monitor the beach evolution in response to the intervention [6–8].

The shoreline, defined as the dynamic interface between land and ocean, is the most common monitoring coastal indicator in morphodynamic studies [9–11]. As the shoreline constantly changes due to cross-shore and alongshore sediment movement in the littoral zone, as well as because of the dynamic nature of water levels, it reveals helpful information on beach variation, being useful for coastal zone monitoring after nourishment interventions [12–15].

Over the last three decades, shoreline evolution has mostly been studied by remote sensing techniques, namely satellite and shore-based video stations. Satellite imaging is a suitable tool for updating shoreline maps, since they provide long-term observations of coastline changes on regional and national scales [16–21]. Nevertheless, shoreline from satellite can be retrieved with low time frequency, and tidal information at the time of image acquisition is often missing.

In contrast with satellite images, coastal video monitoring provides high-frequency, high-quality and continuous images of the nearshore area [22,23]. Shore-based video systems are composed by optical devices installed on an elevated position observing the nearshore area. The continuous acquisition of images allows the observation of the dynamic changes of the nearshore, including the shoreline, in order to build a long-term dataset for a detailed description of coastal morphodynamic evolution [22–24].

Coastal video monitoring operates specific optical products: Timex and Variance images. Timex are digitally averaged images, collected over a period of sampling (generally 10 min), which smooth out the moving features [25–27]. Variance images are built instead as the standard deviation of the sampled period, showing with bright pixels the areas with larger temporal variability, and with dark pixels the unvaried areas [27–29].

Due to their properties, Timex and Variance images have been widely used to detect the shoreline, as they smooth the water movement on the beach face [12,30–41]. However, he number of video imagery applications to monitor coastal nourishment is scarce [42–45]. Harley et al. [43] analyzed shoreline evolution in response to a gravel beach nourishment on the Adriatic Italian coast, with a focus on coastline rotation and recession in response to storm events. Brignone et al. [45] aimed to test the feasibility of a webcam to evaluate the efficiency of a nourishment project carried out on a gravel beach at the Tyrrhenian coast (Italy). Ojeda and Guillén [44] studied the sandy nourishment in two artificially embayed beaches on the Spanish Mediterranean coast, while Elko et al. [42] applied video imagery to monitor nourishment evolution southern an inlet split in the west coast of Florida, facing the Gulf of Mexico. All these works quantified the effectiveness of nourishment works in extending the beach area, however they were conducted at low-energy and micro-tidal systems, where storms play a major role in shaping the coastline.

The main objective of this work was to monitor the nourishment works and to analyze the preliminary shoreline response on Tarquínio-Paraiso beach, a sandy shore in the high-energy meso-tidal Portuguese Atlantic coast. A shore-based video monitoring station was installed on a hotel roof-top and has been storing high-resolution images of the study site. The collected videos during the nourishment works were visually screened to draw up the nourishment calendar, spatially and temporally locating the three phases of works. The rectified Timex and Variance images were used to detect the shoreline manually and automatically, respectively, over the monitored period of six months. The built shoreline dataset was analyzed to quantify the evolution of the emerged beach extent in response to nourishment, completed in summer, and further to winter high-energy events.

This study constitutes the first analysis of shoreline variation at the study site. Moreover, the automated shoreline detection and a video-derived breaking wave height technique were tested to set the ground for an automated video-based integrated system capable of describing hydro-and morphodynamics of the Tarquínio-Paraíso beach.

## 2. Methods

### 2.1. Study Site and Video Station

Costa de Caparica is a sandy stretch located on the south margin of the Tagus river estuary, on the central Portuguese coast (Figure 1a). This area represents the main site for coastal recreational activities of the Lisbon and Setúbal regions [46]. The area has experienced coastline retreat of about 200 m in the last 50 years, resulting in more vulnerability of the urban front to severe storm events [47–49]. In order to protect the urban front, nourishment operations have been performed at Costa de Caparica on an yearly base since 2016 [46]. In 2019, the nourishment activities took place between the 13 August and 24 September. A total sand volume of 1,000,000 m$^3$ was distributed on the seven beaches of Costa de Caparica, between the northernmost São João beach and the southernmost Nova Praia beach [46].

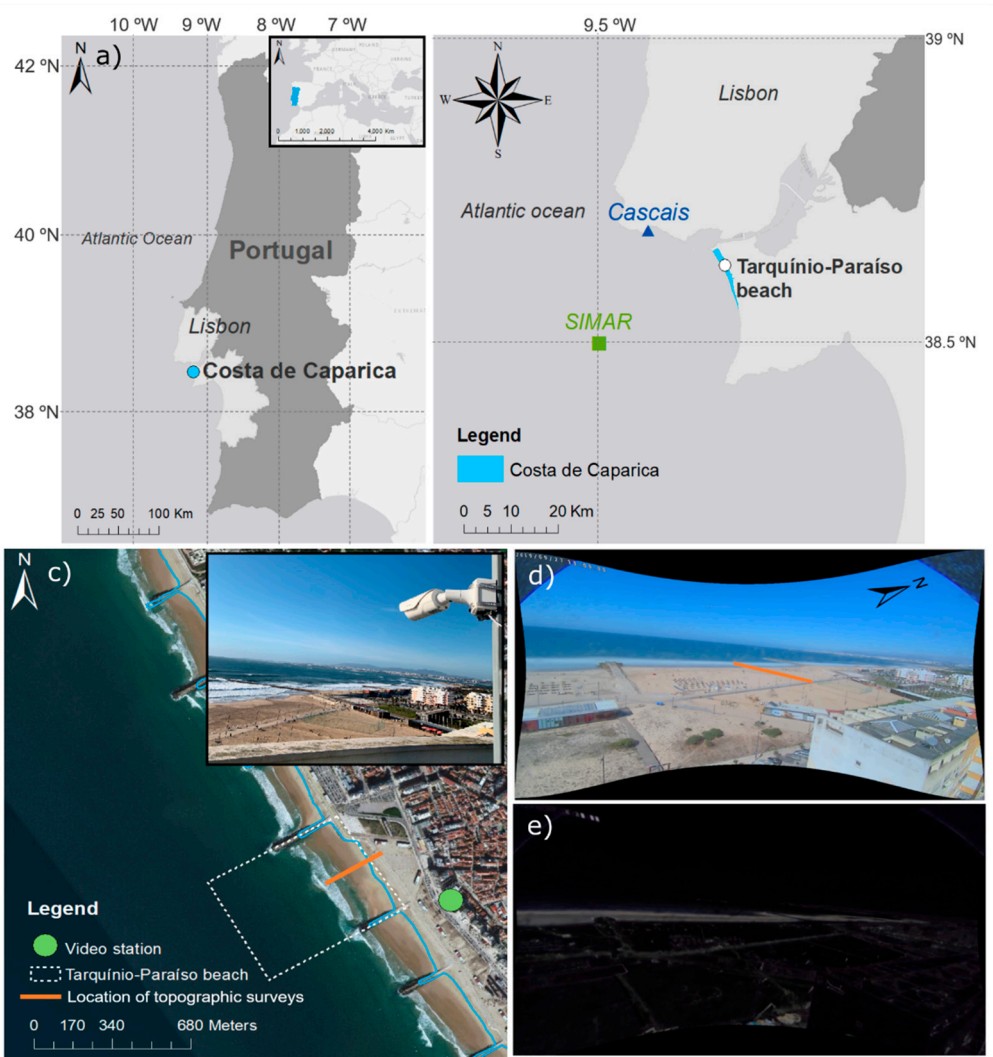

**Figure 1.** Study site map. (**a**) Location of Costa de Caparica; (**b**) location of Tarquínio-Paraíso beach (circle), Cascais tide gauge (triangle), and SIMAR point (square); (**c**) location of the video-station (in green) and of the cross-shore profile surveyed over the monitored period (orange, see Section 3.2); (**d**) Timex image and (**e**) Variance image.

The study site is the Tarquínio-Paraíso beach (38°38'30.3" N, 9°14'20.5" W), the fourth urban beach in Costa de Caparica (Figure 1). The beach extends for about 390 m along the shore with NW-SE orientation, and it is limited sideways by two groins, and landward by a beach wall. The site is characterized by a mesotidal tidal regime, where the average tide amplitude is 2.10 m, and its maximum elevation is 4 m. The wave regime has an average significant height of 2 m, and periods between 7 and 15 s, predominantly coming from Northwest [27,50]. The nourishment activities occurred between 26 August and 4 September 2019, with about 140,000 m$^3$ of sand placed on the shore [46]. The grain size of the nourished sand ($D_{50}$ = 0.55 mm) matched the characteristics of the native sand [46].

To characterize the topographic changes, a cross-shore beach profile was surveyed with RTK-GPS instrumentation prior and after the nourishment works over six months (Figure 1c,d).

Tidal data was obtained by the tide gauge of Cascais (38.69° N, 9.42° W) (ftp://ftp.dgterritorio. pt/Maregrafos/Cascais_radar/2019/), while Hindcast wave and wind hourly data were provided by Puertos del Estado (http://www.puertos.es), a state-owned Spanish company with headquarters in Spain, at the most representative SIMAR point (38.50° N, 9.50° N) (Figure 1b).

A video-monitoring system, comprising an Internet Protocol Vivotek IB9365-HT camera, was installed on a hotel rooftop at 90 m above mean sea level (MSL), looking at the Tarquínio-Paraíso beach, on 30 July 2019 (Figure 1c). The system has been acquiring video-images at 2 Hz, continuously during daylight hours, recording 15 h per day. In this work, the dataset consisted of 126 days, as video data was lost between 14 November 2019 and 10 January 2020.

The video imagery dataset was corrected by the lens-inducted distortions following the Bouguet procedure [51]. Subsequently, 10-min Timex (Figure 1d) and Variance (Figure 1e) images were produced and rectified at the tidal level corresponding to the acquisition time. The rectification procedure was based on collinearity equations [52,53].

## 2.2. Beach Nourishment Monitoring

The nourishment works started dredging sand from the bottom of the southern bar channel of the Lisbon port, at about a 17 m depth.

The dredged sand was transported through submerged metal pipes from the drag suction dredge to the shore, where it was pumped on the subaerial beach and repositioned by caterpillars. This nourishment procedure was considered as the most efficient way to avoid the disturbance of fishing boats in the nearshore, and to reduce the environmental impact on the beach [46].

From the collected videos, three main nourishment phases were distinguished (Figure 2):

1. The pipe-laying phase, when tubes coming from the dredge boat to the shore were placed (and moved) on the beach;
2. The sand injection phase, which consisted in the actual nourishment, when the sand was pumped on the beach;
3. The re-distribution phase, when caterpillars redistributed the dredged sand on the beach.

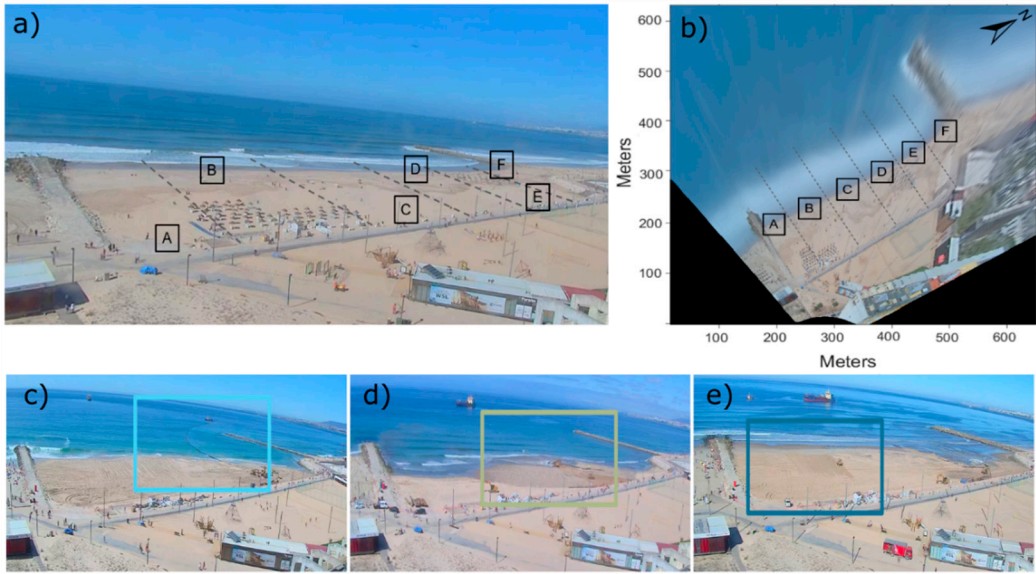

**Figure 2.** Different nourishment zones and nourishment phases at the Tarquínio-Paraíso beach. (**a**) Nourishment zones; (**b**) nourishment zones displayed on rectified image; (**c**) pipe-laying phase; (**d**) sand injection phase; and (**e**) re-distribution phase. The different rectangles indicate the specific area at the beach where each phase is taking place.

Based on the aforementioned definitions, it was possible to elaborate the nourishment map/calendar, identifying the days and zones where the distinct nourishment phases took place. For a regular elaboration, the beach was divided in six different sectors (Figure 2a,b), each representing an approximate longshore extent of 70 m

## 2.3. Shoreline Detection

In order to describe the shoreline variation over the monitored period, a series of images was selected to apply manual and automated shoreline detection. Considering the video dataset, a first screening excluded poor quality images deteriorated by dust or rain drops on the lens, and those images acquired during adverse weather conditions or affected by sun glitter. A second criterion was to reject images with a crowded beach, as the presence of people on the shore may affect shoreline detection. Finally, among the remaining available video data, images with tidal level corresponding to the MSL were selected (i.e., tidal level within ± 0.02 m), following the procedure proposed by Chang et al. [54] and Harley et al. [43]. This approach allowed to minimize the influence of tidal variability on the resulting shoreline position, and to compare shoreline positions taken at the same sea level.

### 2.3.1. Manual Shoreline Detection

The manual shoreline detection procedure was performed on rectified Timex in Matlab environment. To make the procedure regular, a series of 37 parallel transects, with an offset of about 10 m and perpendicular to the beach wall, were superimposed on the image (Figure 3). The detection process consisted in manually marking the limit between water and dry sand at each transect, interpreted as such by the operator.

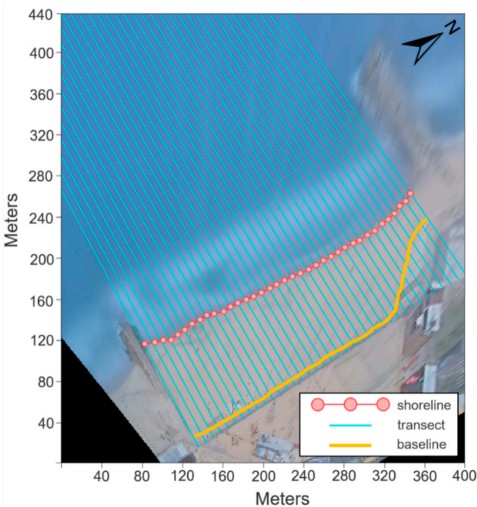

**Figure 3.** Manual shoreline detection procedure performed in Matlab environment. The red circles indicate the point where each transect intercepts the limit between water and dry sand, and the whole set of 37 points represent the shoreline position. The yellow line indicates the baseline, which corresponds to the position of the beach wall, against which the shoreline position is measured.

The baseline chosen corresponded to the beach wall backing the beach. Therefore, the actual cross-shore extent of the emerged beach was found by subtracting the baseline, at each transect, from the shoreline position. The shoreline variation analysis consisted in comparing the shorelines detected over the monitored period.

### 2.3.2. Automated Shoreline Detection

Besides manual detection, a dedicated algorithm was implemented to automatically mark the shoreline. For the automated detection, Variance images were chosen as principal sources, and the same 37 transects used for the manual detection were exploited for sampling pixel intensity.

Given $I_{TX}$ the pixel intensity sampled on Timex, and $I_{VAR}$ the pixel intensity sampled on Variance on the transects (see Andriolo et al. [27] for a detailed explanation), the steps undertaken by the Matlab-based detection algorithm were the following:

1.  Masking dry beach. The color ratio Red: Green bands were computed from $I_{TX}$ profile. A conservative value of 1.4 was used to filter out the emerged beach on $I_{VAR}$, similar to the method in Andriolo et al. [27].
2.  Min–Max normalization of the Blue band of $I_{VAR}$. The pixel intensity statistical values of $I_{VAR}$ are transformed to the range 0–1.
3.  Smoothing data. $I_{VAR}$ are smoothed with a moving average window of 10% of the total transect length.
4.  Masking surf zone. We searched the first peak of $I_{VAR}$ seaward the dry beach limit, which identifies the surf–swash zone boundary ($Sw_{min}$, as proven in Andriolo et al. [27]).
5.  Detrending. The mean value of $I_{VAR}$ profile, taken between the dry beach and the swash zone limits, is subtracted from the main vector.
6.  Shoreline detection. The shoreline is identified at the cross-shore location in which $I_{VAR}$ has a null value.

## 3. Results

### 3.1. Hydrodynamics

Figure 4 shows the hydrodynamics during the monitored period. The highest $H_s$ values and the most energetic days occurred between November and January, the main event being in December with the maximum $H_s$ = 7 m. The wave direction varied between 225° and 350°, with predominant Northwest (NW) direction.

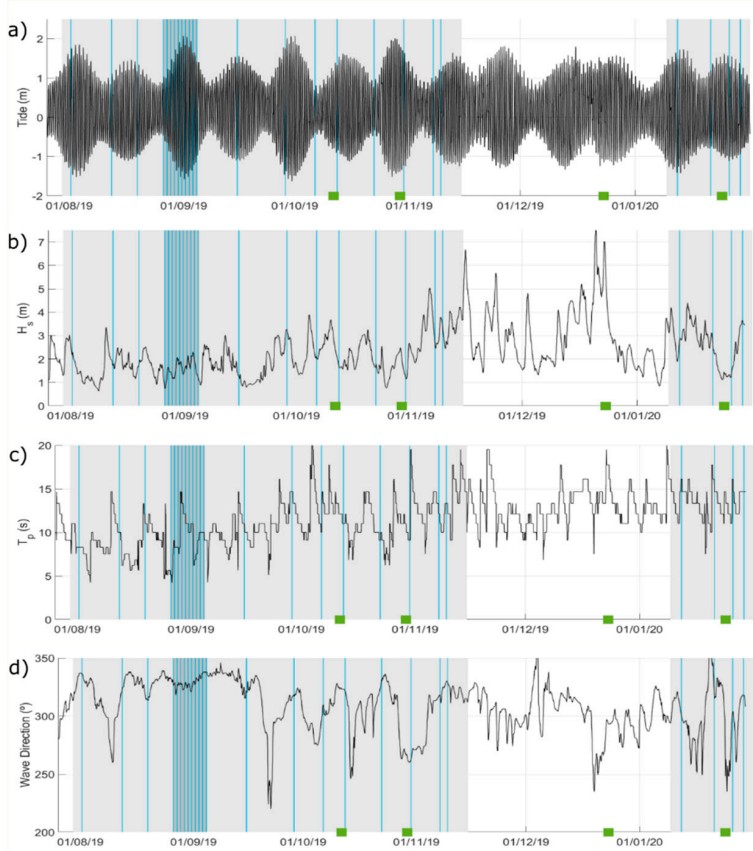

**Figure 4.** Hydrodynamics of the study site. (**a**) Tide level from Cascais tide gauge; (**b**) significant wave height; (**c**) wave period; and (**d**) wave direction (from Puertos del Estado). Light gray rectangles indicate the video-monitoring period, blue vertical lines indicate the days chosen for shoreline detection, and the green squares represent the days topographic surveys were performed during the monitored period.

Figure 4 also indicates the days chosen for shoreline detection and the days where beach profile surveys were performed. Over the monitored period, the shoreline detection frequency was biweekly. Nonetheless, during the nourishment period, the shoreline was detected daily to describe the evolution of the beach extent. Moreover, in November and January, the shoreline was detected weekly, as high energy episodes were more frequent. The topographic surveys represented were performed after the nourishment works over the monitored period to quantify the 3D evolution of the beach profile.

*3.2. Beach Profile Evolution*

Figure 5 shows the topographic evolution of the chosen beach profile (see Figure 1c,d for location on the beach). The profile of 18 July 2019 corresponded to the beach configuration prior to the nourishment and the monitoring period (slope of 0.035), while the profiles acquired in October were the first-available beach configuration after the nourishment.

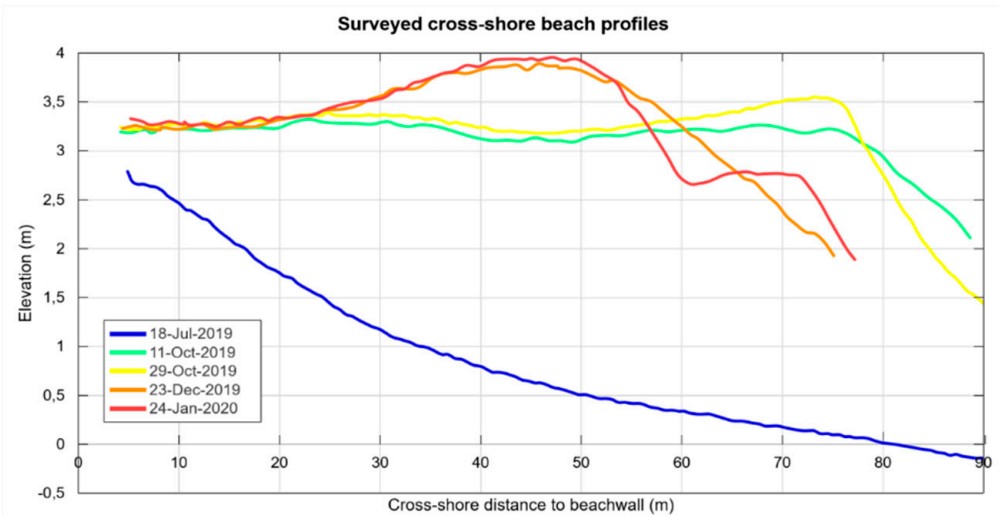

**Figure 5.** Surveyed cross-shore beach profiles of the Tarquínio-Paraíso beach. The elevation of each profile is referred to mean sea level (MSL).

At 60 m location, the slope was inferior to 0.01, showing little variation in elevation between the surveys. After 60 m, the profile from 29 October showed an increase in elevation of about 0.30 m, when compared with the previous profile, which could be attributed to the deposition of sediments by strong waves registered between 27 and 29 October 2019.

The profile of December showed a dome of sand accumulation of 1 m between the 25 and 60 m locations. There was an increase in the frictional forces between the high energy waves, because of the high energy episodes in December (Figure 4b) and the beach surface, resulting in more sediment deposition in this area. This dome represented an increase of approximately 27 m$^2$/m in sand volume, in comparison with October, for that same area. Moreover, on the January profile, while there were no high energy events comparable with the ones from December, according to Figure 4b and the available video data, the sediment accumulation dome prevailed. After December, some of the accumulated sediments started being mobilized by wave action, resulting in a secondary berm, at 60 m location.

Considering the surveyed cross-sections (Figure 5), the unit beach volumes were estimated as the volume of sand contained in a unit length of beach, computing the integral of the beach profiles. Before nourishment, the unit volume was about 83 m$^2$/m, and increased to 263 m$^2$/m after the nourishment works. By the end of the monitored period, in January, the unit volume decreased to 235 m$^2$/m. The estimation of the total volume of the beach accounted for the longshore extension of 390 m. The total sand volume was approximately 33,000 m$^3$ prior to the nourishment, over 100,000 m$^3$ in October, and about 90,000 m$^3$ in January.

### 3.3. Beach Nourishment

Nourishment Calendar

The nourishment calendar is shown on Figure 6. The nourishment activities started from the central zone of the beach (zones B and C), and then moved to zones A and D. Finally, on the last days of the nourishment period, the works took place on zones E and F, located on the northern sector of the beach.

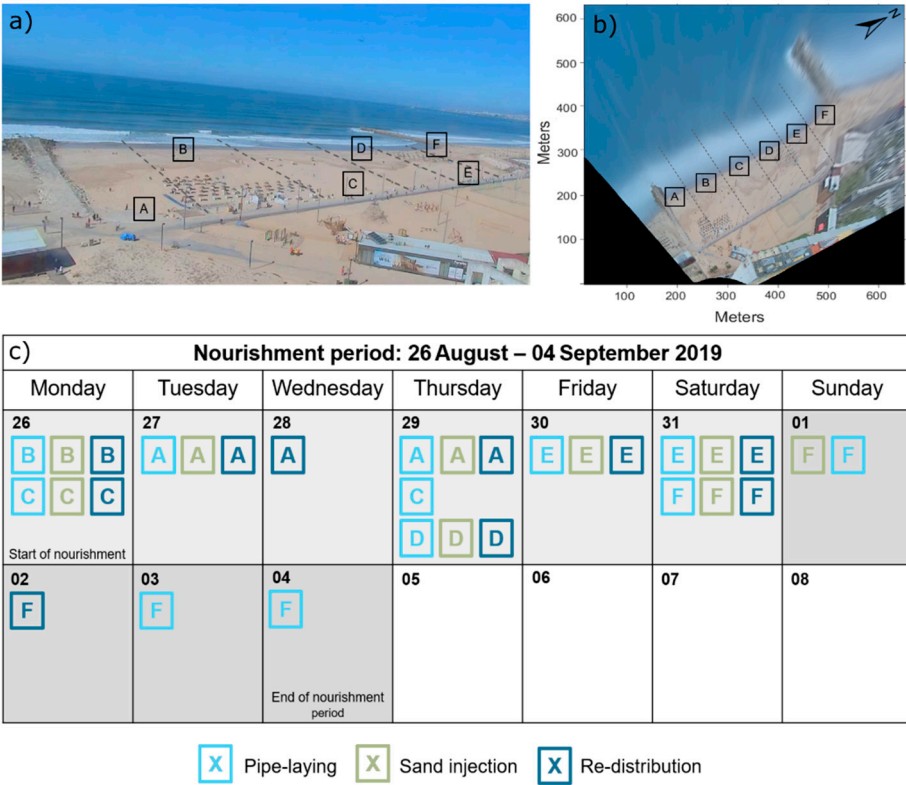

**Figure 6.** The beach nourishment works calendar for the nourishment period. (**a**) Nourishment zones; (**b**) nourishment zones displayed on rectified image; (**c**) the nourishment calendar, indicating the zones where each phase took place, on each day.

The three phases were conducted and completed in each sector (or sectors, if they were being conducted on different zones, at the same time), before moving to the next one. On each different zone, the works were completed in 1–2 days, the exception being zone F. On the last two days of nourishment, the sand injection and re-distribution phases had already ceased, only the pipe-laying phase was still occurring on zone F, as they were already preparing the tubes for the nourishment works on the next beach.

### 3.4. Shoreline Variability

The shoreline variation registered over the monitored period is shown in Figure 7. The initial emerged beach extension was inferior to 50 m. After the nourishment works, the shoreline advanced between 80 and 100 m, for the entire longshore extension, resulting in an emerged beach width of 130 m on average. Between November and January, due to high energy wave events, the shoreline retreated over 30 m. It was not possible, however, to analyze the shoreline position evolution during this phase since it coincided with the data loss period. On the last monitored days, the emerged beach width was about 30 m larger than in August, on average.

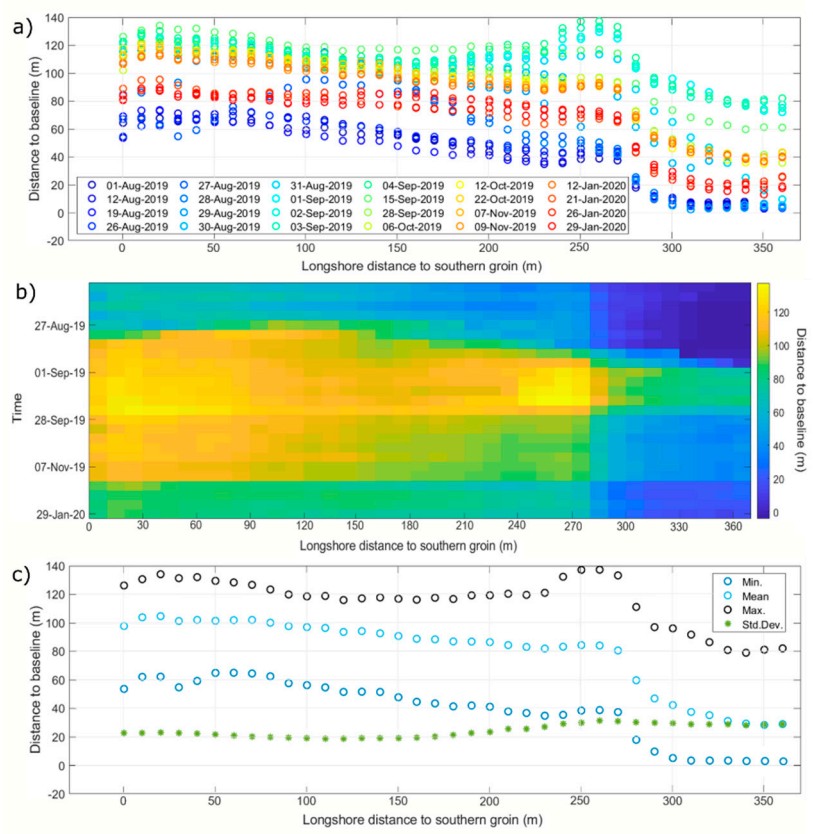

**Figure 7.** Shoreline position evolution. (**a**) Shoreline evolution through the monitoring period (August–January); (**b**) time-space image displaying the shoreline evolution: rows indicate the date of Timex used for the shoreline detection, columns represent the longshore distance (space) from the southern limit of the beach, color represents the beach width; (**c**) the minimum, mean, maximum, and standard deviation values of distance to baseline/beach extension, during the monitored period.

In general, the southern sector of the beach registered the largest emerged beach width, while the northern sector was the area most affected by erosion events, with significant shorter mean beach extent (Figure 7c). The standard deviation values reached its peak around 250 m from the southern groin (Figure 7c), as supported by the significant increase in beach width registered for that same area, during the nourishment (Figure 7b).

Manual vs. Automated Shoreline Detection

The relation between manual and automated shoreline detection is shown on Figure 8. The automated results were satisfactory, with a median disparity of 5 m and an averaged RMSE of 10 m, when compared with the manual technique. Considering the transects, the southern sector was more difficult to retrieve automatically in comparison with the northern sector. This may be due to the lower resolution of the rectified images on the southern sector since it was the furthest from the camera. The detection on the northern limit profiles was also affected by some uncertainties, perhaps due to the sampled transects being closer to the groin on rectified images. Here, the sampling algorithm may have been affected by shadow and wave diffraction generated by the groin. Considering the automated shorelines detected over time, it is of interest to note that later shorelines (November 2019 and January 2020) were detected slightly seaward when compared to the manual shorelines (about 5 m on average). During more energetic days, it was more difficult to visually identify the limit between water and dry sand on Timex, as swash excursion was larger and more irregular. In this regard, Variance images are more appropriate to correctly identify the averaged swash.

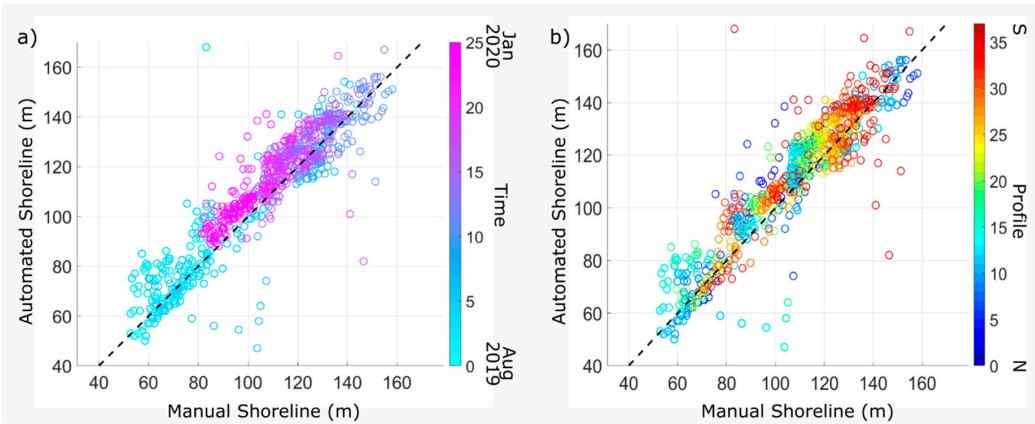

**Figure 8.** Manual and automated shoreline detection. (**a**) Comparison based on shoreline date; (**b**) comparison based on sampled transect. Dashed black line indicates identity.

## 4. Discussion

In this work, the shore-based video imagery technique confirmed its versatility and suitability to monitor nourishment works and study coastal processes. The beach nourishment calendar (Section 3.3) provided a spatial and temporal overview of the procedures, by identifying the different zones and phases of the nourishment works. The example shown in Figure 6 was useful to understand the nourishment dynamics and provides a useful costal management tool for supervising the works on the beach. It is worth noting that in the works monitored by Harley et al. [43] on the low-energy Adriatic coast, the sediment (mixed gravel) was deposited by the dredger directly on the shore and redistributed by wave action. Since the dredger needed to repeat the action of digging, transporting and releasing the material several times, nourishment works lasted about 22 days for an embayed beach of 1 km. In this study case, the sand was transported through pipes from the dredgers to the shore and reallocated by caterpillars on the emerged beach. Although the use of machineries may have a negative ecological impact on the beach, this allowed us to fasten the works and complete the nourishment of the Tarquínio-Paraíso beach (390 m) in about one week.

The use of continuous video monitoring allowed the detailed quantification of the success of the nourishment project in increasing the width and area of the emerged beach (Section 3.4). However, the analysis was limited to shoreline advance and retreat, as video imagery did not permit the evaluation of the variation of beach volume. It was not possible to distinguish the influence of longshore and cross-shore transports in sediment dynamics. On one hand, the beach orientation (NW-SE) in relation to the predominant wave direction (NW) indicate that longshore current may have a significant impact in shaping the beach, in particular in the downdrift side of northern groin. On the other hand, the limited length of the beach (390 m) suggests that cross-shore sediment transport may play the major role in beach erosion and accretion. Similar constraints related to shoreline-based studies were highlighted by the other authors that used video monitoring technique to evaluate nourishment works [43,44], although these analysis regarded shoreline variation at low-energy Adriatic and Mediterranean coast.

The beach profile analysis (Figure 5) has shown that sand was moved up the intertidal area by storms during the energetic winter months, when shoreline analysis indicated shoreline retreat of about 30 m in respect to autumn months. The sand on the emerged beach has likely remained in the beach system though, as coastal processes likely redistributed the sand on the shore and perhaps increasing the dry beach area. In this regard, further work will increase the frequency of beach profile surveys and use a longer video dataset to fully evaluate the efficacy and effectiveness of the nourishment performed in summer 2019.

The coastal video station has been installed with the aim of a long-term monitoring of the coastal evolution of the Tarquínio-Paraíso beach, and it is still operative. In this perspective, the automated shoreline detection proposed in this work (Section 3.4), similar to Emami et al. [55], was shown to

be reliable and returned adequate resolution when compared to the manual detection. At the study site, the use of Variance images for developing the automated algorithm was chosen, as the saturated beach may affect and mislead the shoreline detection on Timex [30,33,37,56,57]. Automated detection also overcomes the subjectivity of manual procedure, since the identification of the water–sand limit is based on the operator interpretation. Nevertheless, the use of Variance is recommended for detecting the shoreline on an unoccupied beach. The presence of people (beachgoers, fishermen, surfers etc.) and moving objects (tractors, quads etc.) are highlighted on Variance as bright pixels like the swash movement exploited to detect the shoreline. Therefore, a crowded beach negatively affects the automated detection [27], whereas on Timex moving effects are smoothed out [22,23].

To improve the morphodynamic analysis, the $Hs_{b,v24}$ method [58] was tested, to estimate the wave height using Timex image (Figure 9). The method is based on the findings that the cross-shore length of the typical time-averaged signature of breaking wave foam on Timex, can be empirically associated with the local water depth at breaking, thus to breaking wave height [58].

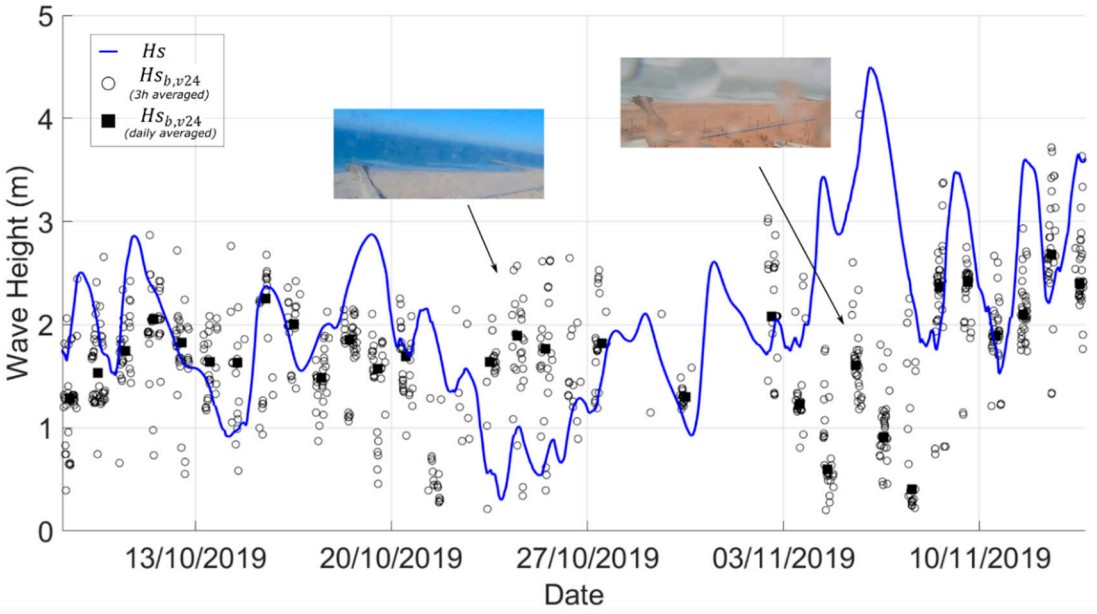

**Figure 9.** Video-derived breaking wave height $Hs_{b,v24}$ [58]. Insets show two examples of images with dirty lens, rain drops and presence of rip currents in the nearshore, which led to poor results.

A total amount of 35 days, covered by about 3000 rectified Timex images, were processed to sample the wave breaking bright pixel pattern over the nearshore bar and eventually applying $Hs_{b,v24}$ method to retrieve the estimation of breaking wave height for each 10-min Timex. The video-derived results were 3-h and daily averaged to be compared to 3-h offshore wave data available (Section 3.1). Overall, the video-derived wave height had a correlation coefficient of 0.35 with the offshore data. Although the propagation of offshore data, considering period and direction, would have provided a more reliable evaluation of video-results, at this stage it was not considered. The measures were negatively affected by rain drops and dust on the lens, presence of mist and low visibility on the beach (Figure 9). In addition, the fully automated methodology returned a poor description of breaking pattern when rip currents were present on the nearshore, as already pointed out by Andriolo et al. [58]. Therefore, an automated algorithm to discard the low-quality images will be developed.

The preliminary test for video-derived breaking wave height is intended to be combined with the automated shoreline detection to build a video-based integrated system that will fully describe the nearshore hydrodynamics and morphology. Besides the breaking wave height and shoreline, the system is expected to provide nearshore bathymetry and hydrodynamics [27,33,59–62], wave runup [63–65] and intertidal topography [33,57].

## 5. Conclusions

This work evaluated the preliminary shoreline response to a sandy nourishment carried out in the wave-dominated Portuguese coast during summer 2019. In addition, this work constitutes the first high-frequency and high-resolution shoreline changes description at the study site, the Tarquínio-Paraiso beach.

Imagery collected by a coastal video monitoring station was used to draw up the nourishment calendar, and to quantify the emerged beach variation over six-months. Timex and Variance images were used to detect the shoreline, by manual and automated techniques, respectively.

The nourishment calendar showed that the phases of the works, namely tubing, injection and re-distribution of sand, started from the center of the beach and ended at the northern sector. For each of the six areas chosen to divide the beach, the three phases were usually completed in one day. Overall, the nourishment of the beach, measuring 390 m longshore, was finalized in ten days.

The nourishment works increased the cross-shore beach extension of 90 m on average. During high energy events in autumn, the shoreline retreated about 50 m. After six months, the emerged beach extent was about 30 m larger than it was prior to the nourishment, with a similar longshore configuration. The preliminary analysis of the beach response to the nourishment highlighted that the northern beach sector is the most vulnerable, with rapid beach extension decrease, as it is the downdrift side of the groin. Overall, the beach extent increased by the sand supply, preventing the usual flooding occurrences at Costa de Caparica, during high-energetic events, until the end of January.

The automated shoreline detection and video-derived breaking wave height were successfully tested and showed promising results, setting the ground for an automated video-based integrated system that will fully describe the nearshore hydrodynamics and morphology at the study site.

**Author Contributions:** Conceptualization, C.J.S., U.A. and J.C.F.; methodology, C.J.S., U.A.; software, C.J.S., U.A.; validation, C.J.S., U.A. and J.C.F.; formal analysis, C.J.S., U.A. and J.C.F.; investigation, C.J.S.; resources, C.J.S., U.A. and J.C.F.; data curation, C.J.S., U.A.; writing—Original draft preparation, C.J.S.; writing—Review and editing, C.J.S., U.A. and J.C.F.; visualization, C.J.S.; supervision, U.A., J.C.F.; project administration, U.A., J.C.F.; funding acquisition, U.A., J.C.F. All authors have read and agreed to the published version of the manuscript.

**Funding:** C.J.S. is supported by RISCO—Center for Environmental Risk Assessment and Management and Civil Protection (FCT NOVA—NOVA University of Lisbon); U.A. is supported by the project UAS4Litter (PTDC/EAM-REM/30324/2017) funded by the Portuguese Foundation for Science and Technology (FCT) and by UIDB/00308/2020. The authors also acknowledge the financial help of the FCT projects To-SEAlert (PTDC/EAM-OCE/31207/2017), BSAFE4SEA (PTDC/ECI-EGC/31090/2017), the MARE—Strategic program UID/MAR/04292/2019, and also to the ORLA—Observatory of Coastal Risks (FCT NOVA—Nova University of Lisbon).

**Acknowledgments:** The authors wish to thank Cláudio Macedo Duarte for the fundamental support in the video station installation and management, Ana Nobre Silva and Hugo Teixeira for the help during topographic surveys and image rectification, Rui Taborda for video station installation and project supervision. The authors would also like to acknowledge the Municipality of Almada and the Hotel "TRYP Lisboa Caparica Mar" for the logistical support in the video monitoring system installation.

**Conflicts of Interest:** The authors declare no conflict of interest.

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
