# Peer review of "Shoreline Response to a Sandy Nourishment in a Wave-Dominated Coast Using Video Monitoring"

_water, doi:10.3390/w12061632_

Round 1
Reviewer 1 Report
The paper reports on a long term experiments on the actions of waves on a recenly refurbished beach.
The experiments are well conceived and conducted, and in my opinion they are perticularly interesting since the stretch of coast considered is realtively short (390 m) and well delimited by two groyns- which makes the experiment significant from the point of view of cross-shore sediment transport.
The paper is also well organized and well written, even though a number of points need to be clarified and some extra information must be provided:
- The orientation of the beach must be specified: the graphical indication in fig. 1 b is not adequate. In connection with this, the direction of the wave action with respect to the beach orientation should be briefly discussed. By the look of it, from fig 7, it appears that longshore sediment transport is very limited; could that be explained in terms of wave direction?
- The granulometry of both the native beach and of the nourishment material should be specifed.
- Do the volumes estimated in line 221-226 refer to the volume of the subaerial beach?
- In fig 5, the Authors should explain why all the profiles stop at the nominal 0 elevation? Did you stop at the current shore line or even before that (23rd December and 20 January) ? And how are the volume calculated ?
- fig 7 b is not clear at all: could the Author explain what does the colour represent?
- Paragraph 2.3.2 deserves a better treatment- perhaps a simple schematic picture would help . Or a t least the Authors should explain why the surf-swash boundary is indicated a peak of the Ivar (point 4). And what do they mean by “detrending” Ivar?
There are a few more points to be clarified in the text, and I have marked down in the attached pdf.
I suggest that the Authors revise carefully all these points. After that the paper can be published-in my opinin without any further review
The paper reports on a long term experiments on the actions of waves on a recenly refurbished beach.
The experiments are well conceived and conducted, and in my opinion they are perticularly interesting since the stretch of coast considered is realtively short (390 m) and well delimited by two groyns- which makes the experiment significant from the point of view of cross-shore sediment transport.
The paper is also well organized and well written, even though a number of points need to be clarified and some extra information must be provided:
- The orientation of the beach must be specified: the graphical indication in fig. 1 b is not adequate. In connection with this, the direction of the wave action with respect to the beach orientation should be briefly discussed. By the look of it, from fig 7, it appears that longshore sediment transport is very limited; could that be explained in terms of wave direction?
- The granulometry of both the native beach and of the nourishment material should be specifed.
- Do the volumes estimated in line 221-226 refer to the volume of the subaerial beach?
- In fig 5, the Authors should explain why all the profiles stop at the nominal 0 elevation? Did you stop at the current shore line or even before that (23rd December and 20 January) ? And how are the volume calculated ?
- fig 7 b is not clear at all: could the Author explain what does the colour represent?
- Paragraph 2.3.2 deserves a better treatment- perhaps a simple schematic picture would help . Or a t least the Authors should explain why the surf-swash boundary is indicated a peak of the Ivar (point 4). And what do they mean by “detrending” Ivar?
There are a few more points to be clarified in the text, and I have marked down in the attached pdf.
I suggest that the Authors revise carefully all these points. After that the paper can be published-in my opinin without any further review

Reviewer 2 Report
This paper provides results on beach evolution process in response to nourishment and wave forcing using video monitoring data, which may be of interest to the readers of this journal. However, the manuscript does not properly reveal the novelty of the works/findings that are distinguished from previous researches, which makes me difficult to find the value of this paper at this moment.
Therefore, I suggest the authors to revise the paper before continuing further review.
1) Please add comments to focus the importance of this work in the Introduction by comparing with previous researches. In the Introduction, Harley [38] and Ojeda [39]’s works are described. Please compare the works of this paper with these previous works, and focus why this works are important.
2) In the Discussion, please focus new outcomes and limitations of this study.
3) Please focus what is the novelty of the automated shoreline detection developed (or adopted) in this study if this is the selling point of this study.
4) In Figure 9, the authors show a result of video-driven breaking height, which is quite interesting to me. However, the procedure and importance of this results are not closely described/explained. Please provide additional information on this process and results if they could be a novel research work of this study.
Round 2
Reviewer 2 Report
The manuscript has been revised reflecting all of my suggestions appropriately.
So, I have no further comments.